# Are Cactus Spines Modified Leaves? Morphological and Anatomical Characterization of Saguaro Seedlings (*Carnegiea gigantea*) with Special Focus on Aerial Organ Primordia

**DOI:** 10.3390/plants13233406

**Published:** 2024-12-04

**Authors:** Cristina Betzabeth Miravel-Gabriel, Ryan Koeth, Nayelli Marsch-Martínez, Tania Hernández-Hernández

**Affiliations:** 1Biotechnology and Biochemistry Department, Centro de Investigación y de Estudios Avanzados (Cinvestav), Irapuato 36824, Guanajuato, Mexico; cristina.miravel@cinvestav.mx; 2School of Life Sciences, Arizona State University, Tempe, AZ 85287, USA; rkoeth@asu.edu; 3Research, Conservation and Collections, Desert Botanical Garden, Phoenix, AZ 85008, USA

**Keywords:** Cactaceae, saguaro, cactus development, aerial organ primordia, spines, areole, tubercle, cactus spine identity, *Carnegiea gigantea*

## Abstract

The reduction of leaves was a key event in the evolution of the succulent syndrome in Cactaceae, evolving from large, photosynthetic leaves in *Pereskia* to nearly suppressed microscopic foliar buds in succulent *Cactoideae*. This leaf reduction was accompanied by the development of spines. Early histological studies, dating back a century, of the shoot apical meristem (SAM) in several species concluded that, in succulent cacti, axillary buds became areoles and leaves transformed into spines. However, these conclusions were based on limited observations, given the challenges of obtaining SAM samples from long-lived, often endangered species. Here, we present a complete study of early aerial organ development in seedlings of the iconic *Carnegiea gigantea* (saguaro), characterizing the different stages of seedling development. We focus on the SAM to track the emergence and development of primordia and aerial organs, closely following the spine development from undifferentiated structures. We demonstrate that young, few-days-old saguaro seedlings provide a valuable model for morpho-anatomical and molecular studies in Cactaceae. We also outline optimal laboratory practices for germinating saguaro seeds and conducting histological studies. Our observations confirm the absence of clear foliar structures and the presence of a distinct type of primordia, hypothesized to be foliar but lacking definitive foliar features. Based on our observations and a review of the literature, we revive the discussion on the ontogenetic origin of spines and propose saguaro seedlings as a promising model for studying the genetic identity of SAM primordia.

## 1. Introduction

Cactaceae is one of the most interesting plant families in terms of their modifications from a ‘regular’ dicotyledonous body plan. Members of this family have adapted to arid and semiarid environments by developing morphological, physiological, metabolic, anatomical, and developmental modifications [1], which together conform their ‘succulent syndrome’. All modifications present in the succulent syndrome are highly interrelated [2]. Although different lineages of succulent plants show variations in the succulent syndrome, several traits are common among all succulents, like the development of large vacuoles, the presence of thick cuticles, low stomatal densities, etc. [1,2,3,4].

Changes in developmental programs that led to organ modifications are perhaps the most dramatic changes in the succulent syndrome, allowing a cascade of other adaptations to evolve. A key element in the succulent syndrome is the reduction or loss of certain organs (leaves or stems) together with an increase in the size of others that store water. The tremendous reduction or loss of leaves, and the total translocation of the photosynthetic functions to the stem allowed the dramatic increase in volume to surface ratios and water storage tissue in succulent cacti [5]. In Cactaceae, the reduction of leaves can be traced gradually through the evolution of the family (Figure 1). Species in the genera *Pereskia* and *Leuenbergeria* are the earliest divergent members within Cactaceae [6]. They lack the typical adaptations most succulent cacti possess, with traits that have been considered as relictual: broad, laminated, thin photosynthetic leaves (C3 photosynthesis), and non-succulent stems, and they inhabit more humid environments [5,6,7,8]. In the Opuntioideae tribe, sister to Cactoideae, most species possess small to minute succulent laminar or terete leaves, enervated, photosynthetic, and often ephemeral [5,9]. In contrast, members of the Cactoideae have dramatically reduced or lost their leaves [5].

The leaf suppression or their suggested modification into other organs such as spines in Cactaceae has intrigued scientists for many years. Studies at the anatomical level started early, with microscopic observations of aerial organ development dating from the 1940’s (for example, see [10,11,12]; as well as [13] and references therein). These early studies, performed on the SAM (shoot apical meristem) of adult plants, lead cactologists to suggest that spines arose by the modification of leaves [1]. However, a gradual loss of the blade in foliar-like structures, hardening of the axis and formation of a spine have not been observed, even in leafy cacti (see [1], p. 34). More recent anatomical studies reported that although reduced, all cacti possess foliar primordia structures, which are microscopic in the case of highly succulent Cactoideae members; and only noticed in studies of the SAM [14]. In 2007, J.D. Mauseth conducted a comprehensive study of these putative foliar structures in the SAMs of mature plants from 147 Cactoideae species (examining 2–4 apices per species). He found these structures in the majority of species, and in some cases identifiable foliar features such as stomata and vascular bundles were present [14]. Based on these observations, Mauseth speculated that the genetic basis for leaf initiation and development are probably present in all cacti ([5,14] and references therein). On the other hand, the lack of any foliar structures, such as venation or stomata, in young spines or foliar-primordia like structures makes current hypothesis about the development or suppression of leaves and the identity of spines inconclusive. Until now, spine homology in the highly succulent Cactoideae members has been determined by relative position and developmental data.

To date, the study of leaf and spine development in Cactoideae members has been severely constrained by the long-life cycles of these species and the challenges associated with dissecting and obtaining sufficient SAM tissue from adult individuals, whether in the field or in living collections. The availability of samples is typically limited, as the SAM, located at the apex of the plant, is often densely covered in spines. Logistical difficulties arise when samples need to be transported from field sites, particularly under harsh environmental conditions. Moreover, a critical concern arises with the collection of SAM tissue, because it may cause significant and irreversible damage, potentially terminating the life of an adult plant, which often requires many years to reach maturity. Under these circumstances, the perspective of conducting studies on the genetic basis of leaf and spine identity in the SAM of cactus appears nearly impossible.

In this study, we analyze the initiation and development of aerial organs in saguaro cactus seedlings (*Carnegiea gigantea*, see Figure 2). Saguaro is an iconic species characterizing the Sonoran Desert in North America. We demonstrate the suitability of seedlings as a model system for studying developmental processes in Cactaceae SAM. We report optimal germination conditions and histological techniques, and for the first time, present a comprehensive morphological characterization of the various stages of germination and seedling growth in saguaro. Additionally, we provide an in-depth anatomical analysis of the SAM during the early days of seedling germination. Unlike previous studies focused on adult cacti, our access to substantial quantities of SAM material enabled detailed tracking of aerial organ development, such as tubercle and spine development and the identification of potential foliar buds at this early stage. This allowed us to establish a chronological framework for aerial organ development. Our findings not only advance the understanding of aerial organ development in Cactaceae but also facilitate laboratory investigations into foliar, tubercle, areole, and spine development. We demonstrate that seedlings as young as 4 to 15 days old already exhibit aerial organs in various developmental stages, laying the groundwork for future molecular studies on developmental processes in this intriguing succulent species.

## 2. Results

### 2.1. Establishment of Optimal Seed Age and Germination Conditions

To explore the use of seedlings as a suitable system to perform anatomical and morphological analyses, we first investigated the optimal seed age and conditions for maximum germination rates. For this, we tested for germination rate and viability changes depending on the years of seed storage. We used batches of seeds collected in different years As a first test, we assessed seed viability using Triphenyl Tetrazolium Chloride (TTC) which is commonly used to indicate cellular respiration. We applied TTC at three different concentrations (1%, 2%, and 3%) in seed batches corresponding to four different collection years (1985, 2002, 2007, 2010). Seeds collected around 39 years ago were not viable anymore (see Appendix A). However, all remaining seed batches (some as old as 18 years old) presented the typical coloration indicating viability. As expected, younger seeds showed the highest levels of viability. Even seeds from 2002 and 2008, which were mor than 10 years old at the time of testing, showed viability and germinated successfully (Appendix A).

Significant differences in germination rates were detected between in vitro and in greenhouse conditions, being higher under in vitro conditions (Appendix A). Germination rates were also different among different year-batches, particularly when germination occurred in greenhouse conditions, while the year of collection did not affect the germination rate considerably under in vitro conditions (Appendix A).

For the in vitro germination tests, the highest germination rates were obtained with the growth chamber set to 24 °C (93% germination rate or GR), followed by 20 °C (79% GR) and 37 °C (63% GR). We found that 37 °C leads to an increased rate of evaporation and loss of moisture of the growth media, ultimately preventing germination after the first week and killing the seedlings after two weeks. The MS concentration showing the highest germination success was 1.1 g/L (94% GR) followed by 4.4 g/L (89% GR) and lastly 2.2 g/L (79% GR). Although not with a significant difference, the best germination rate was obtained when sucrose concentration was 20 g/L (88% GR), followed by 30 g/L (87% GR), and lastly 10 g/L (79% GR). We also tested the application of gibberellic acid (GA) to the medium, to increase germination rates. GA concentrations of 1.72 mg/L and 0.43 mg/L did not yield significant differences in germination (92% and 90% GR respectively), but 0.86 mg/L (84% GR), and 2.58 mg/l (77% GR) produced a reduced GR.

### 2.2. Seedling Development, Emergence, and Growth of Visible Aerial Organs (Morphological Analyses)

In order to evaluate the suitability of saguaro seedlings as a model for anatomical studies, we first followed the earliest days of seedling development. For this, a detailed chronological series of observations and descriptions of morphological changes during germination and growth was conducted on 10 plants per day over a 15-day period (days post-germination, or DPG; see Figure 3A). Special attention was given to the SAM region and the young visible developing aerial organs, which could potentially develop into tubercles, leaves, and spines.

Beyond the initial 15 days, growth observations continued for an additional 30 DPG with a smaller subset of plants, and a few plants were monitored until they reached 1.5 years of age (Figure 3B and Figure 4). Root observations and descriptions were brief, as the primary focus of this study was on the SAM region and aerial organs. All morphological descriptions were based on plants grown under in vitro conditions, as these exhibited more vigorous growth compared to those grown in soil in greenhouse conditions. It is important to note that growth and developmental stages may be delayed in soil or natural field conditions compared to in vitro conditions. Nevertheless, these conditions were optimal to perform the developmental observations in a shorter period of time.

The root was the first visible organ to emerge during germination, that occurred generally 2–3 days after sowing (DAS, Figure 3A). The emerging roots were whitish in color and slightly translucid. Lateral root primordia were clearly visible on the sides of what would later become the primary root. In the following days, the root adopted a typical dicot behavior, presenting one main root with lateral hairs (Figure 3B). In some cases, instead of a single main root, many adventitious roots developed. This type of root growth has been previously reported for other columnar cactus species [15,16].

The cotyledons were visible for the first time by the third DAS or the first DPG. They showed a whitish to slightly green color and a nearly triangular shape. In the subsequent DPG, the cotyledons increased their volume and grew longitudinally (Figure 3A and Figure 5). The color of the cotyledons during the first days varied among plants from intense green to purple. There were no significant changes in the cotyledons after their initial thickening and enlargement, and some remained visible in one-year old and older plants (Figure 3B and Figure 4).

Observations of the SAM in seedlings were performed every day (Figure 5). There were no visible structures in the region between the cotyledons during the first DPG (Figure 5a). By the 2nd DPG, two protuberances, corresponding to primary or tubercle primordia, could be observed (Figure 5a,b). These small structures grew rapidly, and by the 3rd to 5th DPG they became wider at the base (Figure 5c–e). Structures that resemble early spines developed at their apical region. Similar observations were made from the 6th to the 30th DPG (Figure 3, Figure 4 and Figure 5f–p). By the 13th DPG, the cotyledons were considerably separated and the stem started to elongate and grow between them. The stem grew in width and height with areoles starting to appear and rapidly increased in number. Tubercles were visible by eye in two-months-old seedlings. After the tubercles developed fully, rib formation followed as a consequence of tubercle fusion. The initial notable enlargement observed in the first days of development is less evident one month after germination, which may be due to the relative size of the new organs with respect to the total size of the plant.

### 2.3. Aerial Organ Primordia Formation and Growth Through Seedling Development (Histological Observations)

To analyze the formation of aerial organs in the SAM, we made histological sections using seedlings sampled each day for 15 DPG. The two different embedding methods tested (paraffin and methyl methacrylate) yielded satisfactory results. However, developing structures could be better observed when methyl methacrylate was used. Figure 6 shows a schematic synthesis of events occurring in the SAM of seedlings during their first 17 days of development with drawings. We describe these events in detail with the respective histological photographs below (Figure 7, Figure 8 and Figure 9). At sight, the SAM and tubercle or primary primordia appeared to develop and grow at a fast pace, with significant changes during the first 10 DPG. These changes were less evident from around the 12th DPG, probably due to the relative size of new organs to the whole plant. The SAM was not yet visible in 1 DPG seedlings (Figure 6a and Figure 7a). At this stage, the cells of the cotyledons have large vacuoles, and the epidermal cells are difficult to distinguish. However, at the region where the internal face of the cotyledons meets the hypocotyl, the epidermal cells are clearly distinguishable, small, and increase their size in the regions farther from the meeting point (where the SAM should normally be visible, see Figure 6a and Figure 7a). The cells just below the epidermis are also smaller than the cells farther from it. In 2 DPG seedlings, the cotyledons become slightly separated, the SAM appears as a flat dome with a tunic of two strata, and the cells become smaller as they approach the apex (above the meristematic zone, Figure 6b and Figure 7b).

The zonation in the saguaro SAM corresponds to the descriptions available for cactus seedlings [17] and adult plants [10]. By the 3rd and 4th DPG, primordia in the SAM increase their size, showing a round shape with the first two external cell layers well arranged (Figure 7c). Considering the position of the organs present at later stages, these primordia appeared to correspond to primary primordia or incipient tubercles (tubercle primordia), which will later develop areoles in their apex. The SAM comprises a region with small cells under the tunic (central mother cells), which showed two cell layers presenting periclinal divisions exclusively (Figure 7d,e). The most external layer, the tunic, was well defined and presented clear divisions (Figure 7d(A) and Figure 7e). The corpus was well differentiated with the following distinguishable zones: zone of central stem cells or initial zone, composed of small cells present at the top of the dome, under the tunic (Figure 7d(B)); the rib zone (Figure 7d(C)); and finally, the peripheral zone that will produce the future primordia (Figure 7d(D)). The peripheral zone presented small to medium sized cells and apparently no stratification.

By the 4th DPG, primordia enlarge, with the apex cells irregular in size and arrangement, and organ primordia (spines) started developing to the periphery of the SAM, at the base of cotyledons (Figure 7d,e). Primordia continued to emerge and grow in pairs, consecutive to each other, either in a parallel (Figure 6c and Figure 7e) or perpendicular orientation (Figure 7g), with respect to the cotyledons. This is also observed in the images taken from the top of the seedling, in Figure 5. When a primordium emerges perpendicular to the cotyledons, the SAM is not visible in outer sections. This primary or tubercle primordium showed a perfectly defined ovoid body (Figure 7f,g).

In transverse sections, numerous primary primordia (around 4–5) could be observed in a radial disposition around the SAM in 7-day-old seedlings (Figure 8a,b), and these primary primordia increased in size in the subsequent days. As indicated before, they emerge from the SAM in pairs, one pair appearing immediately after the other, but in the opposite disposition, creating an initial cross-shaped phyllotaxy. These first four primary primordia indicate the beginning of the structures (tubercle primordia) that later in time will develop as ribs. The next pair of primordia (the 3rd) appears with a similar organization pattern, close to the place where the first two were formed. The fourth pair appears close to where the second was formed, etc. Due to the difference in position and time of tubercle primordia emergence, the final phyllotaxy corresponds to alternating tubercles with a helicoidal pattern (Figure 4a–g). At much later stages, the tubercles “align” vertically and fuse, forming the ribs (Figure 4k,l). As the plant grows, new sets of tubercles arise, which fuse into new ribs formed as vertically straight rows.

After the 6th DPG, trichomes were clearly visible at the tip of primary primordia (Figure 6g–i and Figure 7h–o). The trichomes are also visible in aerial views of seedlings (Figure 5), observed in between the cotyledons as colorless hairs). At this stage, the connection of vascular bundles from the cotyledons and the areole primordium are visible (Figure 7h). On top of each primary primordium, a region of small cells with meristematic characteristics can be observed, probably corresponding to a young areole (Figure 7e and Figure 8a,b). The formation of spines from this meristematic region can already be observed by the 6th-7th DPG (Figure 6f–g and Figure 7h,i). The spines show a triangular shape and are notably thicker than the trichomes that developed previously (Figure 7h). Spine primordia increase in size and number by day 7 (Figure 6g, Figure 7j and Figure 8a,b), and grow rapidly the next days, becoming visible by eye (Figure 5). The spines developing on the oldest primary primordia appeared to be completely formed by the 10th to 13th DPG (Figure 6l–o, Figure 7l–o and Figure 8d,f–i). Fully formed spines mature progressively over the following days (Figure 8h,i), and afterwards the spines show small protuberances arranged along them, resembling scales (Figure 8h,i). No stomata, guard cells, hypodermis, or chlorenchyma are observed in spines at any time, and they developed as radialized structures from the beginning. We identified two zones in young spines: (1) the basal zone (Figure 7n and Figure 8h,i: darker area at the base), that can be described as a mass of libriform, non-lignified cells growing in line towards the exterior with no vascular tissues or procambium; and (2) the elongation zone, above the basal zone, characterized by sclereid-like epidermis cells that present many scales (Figure 7l–n and Figure 8h,i). While no vasculature was observed in developing spines, developing vasculature tissues possibly irrigating the primordia are present (Figure 7o, Figure 8d and Appendix A).

Although not frequently, and not in all SAMs sampled, we were able to observe microscopic protuberances emerging from the upper portion of developing tubercles with a distinct development from primary or spine primordia. We interpreted those structures as foliar-like primordia or “microleaf-like” structures. They appear to be small protuberances of cells rising at the base of spine primordia that will form the areola, just below the first spines (Figure 9). These structures lacked stomata or conductive tissue but presented mesophyll cells. The vascular tissue in the hypocotyl or early stem of the seedlings is visible and showed a primary protoxylem consisting of 7–8 vessels with helical walls separated from each other by parenchyma cells (Appendix A). These vessels bifurcate as they approach the SAM. Some of them irrigate the cotyledons and others continue towards the apical meristem and primary primordia, connecting to the sides of the base of primordia (Figure 7k,o and Appendix A).

## 3. Discussion

Over evolutionary time, plants exhibiting the succulent syndrome have modified their organs, tissues, and metabolic pathways to adapt to water scarcity. Key modifications include the reduction or loss of certain organs and tissues, coupled with the enlargement of others. In cacti, one of the most significant evolutionary changes was the thickening of the stem, the reduction of leaves and the development of spines, likely to provide stem shading and protect the water reservoir from herbivory. This transformation coincided with the transition of photosynthesis to the stem and the development of a three-dimensional vascular network in the cortex [1,5,14,18]. Despite these insights, the ontogenetic identity of structures like tubercles, ribs, and spines is still not fully understood, and developmental studies of their early formation could help bring greater clarity. Early studies revealed the most current and exiting knowledge, based on observations of adult plants. For example, Freeman’s (1970) work on leaf and spine ontogeny in *Opuntia*, ref. [19] and Boke’s extensive contributions on leaf development in *Opuntia* [10,11] and *Pereskia* [20]. These studies primarily focus on anatomical observations of the shoot apical meristem (SAM) or areolar meristems in mature plants, where tissue availability is limited and extraction challenging. In contrast, seedlings offer a practical and replicable model for studying early development in the SAM. In this study, we propose using saguaro seedlings as a model system to investigate the development of aerial organs in the SAM of succulent cacti, shedding light on the evolution of the succulent syndrome in this important group of xerophytic plants.

Before proceeding to perform morphological and anatomical observations, we established the best protocols and procedures to germinate saguaro seeds in laboratory conditions. Germination rates were notably successful, even with seeds that were approximately 20 years old. However, seeds collected and stored in 1985 (39 years ago) had lost viability. Some of our seed batches exhibited very low GR or no viability at all, despite being relatively recent. We noticed that these seeds were stored at room temperature, in contrast to the remaining seeds, which were stored at −18 °C to −20 °C. Previous studies have demonstrated that storage at room temperature significantly decreases or eliminates GR in various cacti species [21], with low temperatures being ideal for seed preservation [22]. According to our findings, the optimal conditions for germinating saguaro cactus seeds to perform anatomical studies and histological preparations involve using MS media at a concentration of 1.1 g/L with a sucrose concentration of 20–30 g/L and GA ranging from 0.43–1.72 mg/L. The ideal temperature, if a growth chamber is available, is 24 °C. However, if germinated at room temperature and temperatures are higher, careful monitoring of media moisture is essential, and media volumes should be adjusted as necessary to counteract evaporation and prevent desiccation.

Once the adequate protocols to germinate and grow saguaro seedlings were established, we performed a detailed morphological characterization of seedling growth and anatomical observations of their SAM using optoelectronic and standard light microscopy. There is a scarcity of morphological and anatomical studies of aerial organ development (such as tubercles, leaves, and spines) during the early stages of development in cacti, particularly in recent decades (e.g., [23]). Although SAMs in seedlings of several cacti species have been investigated before, those studies focused on the development of zonation in the SAM [17] as well as the variation in apical meristem size, type of zonation present, and the variability of the sizes of the zones [13,14], and did not report an ontogenetic series of aerial organ development as we do here. Therefore, our aim was to perform an ontogenetic series for the SAM and developing aerial organs of saguaro seedlings in order to describe the early tubercle, spine and possible leaf primordia development chronologically. To achieve this, SAM preparations of freshly sprouted plants (0–15 days after germination) were analyzed.

In some *Opuntia* species, once dormancy is broken and germination is complete (when the seed coat is ruptured), seedlings suffer rapid morphological changes during the first days after germination [24]. In saguaro, once roots emerged, they also showed fast growth during the first days after germination (15 days). However, after this period, morphological changes were less notorious, possibly because at this stage growth was concentrated on the root [25]. Since our study was focused on aerial organs, we did not perform detailed observations on the root system.

Our morphological analysis of saguaro seedling growth indicated that most morphological changes in the SAM and aerial organs are evident within the first few days after germination. We closely monitored tubercle, spine, and putative leaf development in the shoot apical meristem (SAM), both at the morphological and anatomical level. Following a notable thickening of the central portion of the seedlings, spines emerged as the first aerial structures visible to the naked eye between days 9 and 12 post-germination (see Figure 3 and Figure 5). These spines underwent rapid changes during development, consistent with observations in *Stenocereus queretaroensis*, where the first areoles appear one week after germination [26]. In *Carnegiea gigantea*, spines emerged from a dense central mass of trichomes. However, no foliar primordia or leaves were observed at any stage of development, either visually or under a dissecting microscope, confirming that if foliar primordia are present, they are either microscopic or ephemeral [14,27]. The first tubercle or primary primordium emerged from the peripheral zone of the SAM and was visible as early as the 2nd to 3rd DPG, similarly to what was previously reported for *Opuntia basilaris* where the first primordium was seen 3 days after moistening seeds [28]. In the case of saguaro, the first recognizable structures observed in the primordia after germination were numerous trichomes that emerged on the upper part of the tubercle (Figure 5). Visible spines appeared two days later. Spines grew fast and matured acropetally, as has been observed in *Opuntia cylindrica* [11]. Along the spine, small epidermal bumps appeared (Figure 8g–i), but disappeared when the spine matured, approximately at 20 DPG, or when plants were exposed to dry conditions.

Unlike in *Arabidopsis*, where the SAM develops at the torpedo stage during embryogenesis [29], the SAM in saguaro seedlings became visible post-embryonically by the second DPG, though with an unclear zonation. This early-stage SAM development is similar to that reported in *Mammillaria brandegeei* seedlings, where only tunica and central mother cells were present at germination [17]. By the third DPG, complete zonation was evident, resembling the development in *Opuntia basilaris*, where full zonation is established by the second DPG [28]. By the fourth DPG, four distinct zones could be identified within the SAM: the Tunica (T) (Figure 7d(A)), Central Mother Cells (CMC) (Figure 7d(B)), Pith-Rib Meristem (PRM) (Figure 7d(C)), and Peripheral Zone (PZ) (Figure 7d(D)), consistent with previous observations in *Opuntia cylindrica*, *Trichocereus spachianus* [10], and several other species across three different subfamilies [13].

Abundant, non-glandular and multicellular trichomes arose from the tip of the observed primary primordia (areole region) apex, and were very evident by the 6th DPG (Figure 5, Figure 6g, and Figure 7j,l). They grew continuously and never lignified, not even in older plants (15 DPG or 1-year-old plants, see Figure 4). In contrast to spines, trichomes did not present acropetal maturation. Even though vascular tissue was not analyzed in detail in this study, some observations were made. Two main vessels developed from the hypocotyl-root junction towards the SAM, and they bifurcated towards cotyledons, later diverging to the superior part of the tubercle (Appendix A). According to Mauseth [4] some columnar cacti present a dimorphic wood transition, which means that wide-band tracheids (WBT) develop in seedlings during the first months after germination and then switch to producing fibrous wood with no WBTs to support tall and heavy bodies [4]. In saguaro seedlings we observed a protoxylem formed of 6–7 vascular bundles with helicoidal walls (Appendix A), or WBT, similar to previously reported in two months old *Neobuxbaumia multiareolata* seedlings [30].

Even though we report a complete and detailed developmental series of aerial organ development in the SAM or saguaro cactus, spine identity still remains ambiguous. Similar to Mauseth’s description of spines in mature individuals [4,5,31], we confirm that they lack any feature characteristic of leaves, like guard cells, hypodermis, stomata, phloem or xylem. Instead, spines consisted of libriform fibers. However, our observations revealed that spine maturation and lignification occur very early and rapidly during their development.

In addition to spine primordia, and only in a subsample of our SAM longitudinal sections, we were able to identify other structures, that resemble foliar primordia or “microleaf-like” structures (Figure 9), and were similar to the ones described by Mauseth in the SAM of several Cactoideae species [14]. In some succulent cactus species, these microscopic ‘microleaf’ structures elongate and grow in thickness, showing vascular bundles and stomata characteristic of leaves [14]. However, according to our observations, the microleaves in saguaro completely lacked vascular tissue, lamina, or stomata, developing as a distinct bump or tip, remaining small and short, as observed in other species [14]. In agreement with earlier findings [14], we hypothesize that these structures could be reminiscent to leaves, since their development seems to be different from the spine primordia observed next to them in histological preparations (Figure 9). In Cactaceae, the evolutionary reduction of foliar leaves was accompanied by an increased growth of shoot cortex and a loss of the abscission zone [14]. According to Mauseth, the small microleaves that develop in the SAM of derived Cactoideae species can be identified by position criteria, using the axillary buds as a marker for the stem/leaf junction, where leaf primordia are small distinct protrusions and axillary buds form very early right at the junction of the microleaf and the SAM [14]. Here we confirmed this observation, since the microleaves observed were preceded by axillary buds later developing trichomes and spine primordia (Figure 9). The position of saguaro in Cactaceae phylogenies is highly derived [6,32], indicating perhaps a more dramatic suppression of the leaf developmental program in this highly succulent cactus species.

One possible explanation of the very low frequency in which we observed these structures is based on the ideas suggested by Mauseth about the rate of leaf and plastochron production [5]. A plastochron is the time interval among new leaves that are produced at the tip of the plant, and it is used as a way of measuring the age of a plant. For example, according to Mauseth, species included in genus *Ariocarpus*, *Lophophora*, *Pediocactus* and *Sclerocactus* produce one to five leaf primordia per year and belong to a group of plants with a low rate of leaf and plastochron production [5].

At this stage, it is still premature to definitively diagnose whether spines develop in a similar manner to leaf primordia or branches. Nevertheless, the observation of the microleaves invites informed speculation regarding their potential developmental origins (Figure 10). We did not identify anatomical features such as stomata or a primary vascular system that could clarify the identity of either the spines or the microleaves. However, the relative positioning and developmental trajectory of these elements provide valuable insights see also [14]. Based on their positioning, we speculate that tubercles, spines, and microleaf-like structures may have distinct ontogenetic origins. The first structure we observed (tubercle or primary primordium) could correspond to an axillary meristem derived from a suppressed leaf, suggesting that this tubercle primordium may be equivalent to a shoot. Subsequent structures developing atop the tubercle, such as the microleaf-like features, could be remnants of a rudimentary leaf, while spines developing adjacent to these microleaf-like structures could originate as shoots from the axillary meristem.

This hypothesis remains speculative, and alternative interpretations must be considered. To achieve greater certainty, additional observations in saguaro seedlings are required, encompassing a larger sample size and a broader range of developmental stages. In addition, observations of the microleaves in different species and using scanning electron microscopy (SEM) are needed to further confirm our observations. Such studies would enable a more precise determination of these structures’ identity and, if they are indeed primordia, their ontogenetic fate. Furthermore, it would also be interesting to determine their rate of plastochron production. At this stage, molecular studies focusing on the expression of genes involved in leaf development could offer definitive evidence regarding the organ identity of spines and tubercles. Seedlings provide an ideal system for conducting molecular experiments, such as in situ hybridization, targeting genes known to regulate leaf initiation and development. This approach would help elucidate the ontogenetic identity of cactus structures such as tubercles, spines, and leaves, while also shedding light on the evolutionary shifts within genetic regulatory networks that led to the suppression of leaf development [33].

## 4. Materials and Methods

### 4.1. Germination

Optimal germination and growth conditions for saguaro seedlings were analyzed in vitro and in greenhouse conditions. Seeds were donated by the Desert Botanical Garden, Phoenix, AZ, USA, and were collected in 1985, 2008, 2014, 2021, and 2023. In addition, a supplementary batch was donated by F. Molina-Freaner, from Instituto de Ecología UNAM, Campus Hermosillo from specimens in the surrounding Hermosillo, Sonora, Mexico area (seeds collected in 2002, 2007, 2010). As an initial step, we performed a tetrazolium chloride test to detect seed viability in some of our seed batches. Seeds were stored at 3 °C and were only removed as needed. Before germination, seeds were weighed to give a count estimate and then placed in a 1.5 mL Eppendorf tube for sterilization. Seeds were sterilized in a 1:1 ratio of hydrogen peroxide and 70% ethanol for 3 min then washed three times with sterilized distilled water before being plated on MS media. The main stock MS media used consisted of agarose 10 g/L, sucrose 10 g/L, MS 2.2 g/L, and PPM 2 mL/L. This base MS media was adjusted to test for different sucrose concentrations of 20 and 30 g/L and MS concentrations of 1.1 and 4.4 g/L.

We performed a screening to find the most adequate germination temperature, since saguaro cactus naturally germinates in the Sonoran Desert in early fall, characterized by high temperatures. Seeds were germinated in three separate growth chambers with temperatures of 20 °C, 24 °C, and 37 °C respectively, with a 16/8-h light regimen. The rate of germination was tracked weekly and concluded after one month for each of the tests. We tested the effect of different gibberellic acid concentrations (0.43 mg/L, 0.86 mg/L, 1.72 mg/L, and 2.58 mg/L), with the seeds showing the best germination rates, which were from 2023.

For germination in the greenhouse, seeds were sown in a 50-cell seedling tray with soil containing volcanic rock/potting soil (1:1). Trays were covered with a clear plastic dome and transferred to the greenhouse. For the morphological characterization of the germination process and the early stages of aerial development, plants were carefully observed and photographed daily following [34], using a VHX-5000 Keyence microscope (Keyence, Osaka, Japan).

### 4.2. Anatomical Analyses

Two different embedding methods to prepare samples for morphological analyses were tested: (1) Paraffin and (2) Methyl methacrylate (Technovit^®^ 7100, Kulzer Technik, Hanau, Germany). We used the plants grown under in vitro conditions as described before. Germination occurred 3 days after sowing. Four plants were collected each day, starting from the first day post germination, for 15 days. A total of 60 plants were collected per embedding method. For the paraffin embedding method, fixation was performed with FAE (3.7% formaldehyde, 5% glacial acetic acid, and 50% ethanol). Four whole plants were placed in Eppendorf tubes and 5 volumes of freshly prepared FAE were added. Samples were exposed to a 15 min vacuum cycle and were left in the tubes for 24 h at room temperature. After 24 h, the samples were rinsed with tap water and stored in 30% ethanol until the setting was completed. The samples were dehydrated using a Leica TP1020 tissue processor by progressively increasing ethanol concentrations. The starting concentration was 40% ethanol, followed by 50%, 60%, 70%, 80%, 85%, 90%, 96% and 100% ethanol (2×) for 48 h each. Ethanol uptake was improved by shaking the samples. Paraffin embedding was performed in the tissue processor using molten Paraplast Plus ^®^ at 65 °C for 48 h each. After embedding, samples were placed into molds for further processing. Samples were sectioned to 10 μm using a rotary microtome (Leica microsystems, Wetzlar, Germany) with a 10-degree cutting angle. Sets of 6–10 sections were placed on glass slides covered with Haupt’s gelatin adhesive and 4% formalin, surplus formalin was removed and the slides were then placed on a hot plate at 55 °C until dried. Samples were stained with safranin and counterstained with fast green. Before staining, samples were dewaxed with xylol. These processes were carried out simultaneously through a “staining train” that consists of 10 stations, as follows: (1) 10′— Xylol I, (2) 10′—Xylol II, (3) 10′—Xylol III (4) 10′—Ethanol 100%, (5) 10—Ethanol 96%, (6) 10′—Ethanol 70%, (7) 10′—Ethanol 50%, (8) 2 h—Safranin, (9) quick rinse in distilled water, (10) 10″ in fast green. Afterwards, stained slides were covered with drops of synthetic resin and protected with a coverslip. Finally, slides were air-dried for 7 days and observed using a light microscope (DM 750, Leica microsystems, Wetzlar, Germany) coupled with a camera (ICC50 HD, Leica microsystems, Wetzlar, Germany).

For the methyl methacrylate embedding method (Technovit^®^ 7100), we performed fixation similarly to the paraffin method. Collected samples were first fixed in FAE. Then, they were placed under vacuum for 20 min and left 2 h in the FAE solution. After 2 h, the solution was discarded and replaced by 70% ethanol and the samples were left overnight at 4 °C. Dehydration was performed the next day, and samples went through series of increasing ethanol concentration solutions, for 1.5 h in each, at room temperature. The starting concentration was at 70%, followed by 85%, 95% and 100% ethanol. We performed a pre-infiltration step following the manufacturer’s recommendations. The samples were incubated in the pre-infiltration solution (SF10-Basic Solution Technovit^®^ 7100:Ethanol 100% 1:1 *v*/*v*) with 20 min vacuum and then incubated at room temperature for 2 h. Samples were incubated in the infiltration solution (SF10-Basic Solution Technovit 7100+Hardener I) for over five months at 4 °C. After infiltration, samples were placed in molds with the polymerization solution (Infiltration solution + Hardener II) and incubated at 37 °C for 48 h. Samples were transversally and longitudinally sectioned in a rotatory microtome (Leica) in 10 μm thick slices and stained with alcian blue (0.5% pH 3.1) for 15 min and neutral red (0.5% pH 3.1) for further 15 min. Microphotographs were taken in a light microscope (DM 750 Leica) coupled with an ICC50 HD camera (Leica).

## 5. Conclusions

The work presented here lays the foundation for establishing *Carnegiea gigantea* seedlings as a potential system for studying developmental processes in Cactaceae, particularly the modifications that occurred in the programs guiding the development of aerial organs. We demonstrate that as early as a few days after sowing and germination, the seedlings’ SAM already exhibits the emergence of primordia for fundamental organs that later develop into tubercles and spines. This enables the study of the development of these organs in meristematic regions in a more practical and easy way, rather than relying on adult plants that can take several years to mature and, in many cases, such as in columnar cacti, have an inaccessible SAM. Our results allowed us to determine the optimal conditions for germinating and growing saguaro seeds under in vitro conditions, and to establish the methodology to perform anatomical observations within the SAM.

We report the postembryonic characterization of the development of aerial organs. An organized SAM can be observed two days after germination. From this point onwards, the emergence of primordia that give rise to different structures (such as tubercles, trichomes, foliar primordia or structures, and spines) is clearly detected by the fourth day. Moreover, observations of the vascular tissues of developing organs in seedlings can be initiated at this stage. We confirmed the different organs that emerge from the SAM: primary primordia that later become tubercles, and spines that develop at the apex of tubercle primordia. Additionally, in a few samples, we observed other structures that we hypothesize to be foliar primordia or remnants of leaf-like structures, which we refer here as “microleaf-like” structures. Further observations in seedlings of different species, as well as observations with more powerful tools such as SEM are needed to confirm the identity of SAM primordia structures.

Although our results do not definitively confirm or refute the hypothesis that cactus spines are modified leaves, we believe our study is of significant interest because it revives the apparently settled discussion surrounding the identity of these organs. Dr. Mauseth’s research and seminal work, summarized on his website (https://web.biosci.utexas.edu/mauseth/researchoncacti/Spines.htm), explains: “*Mature cactus spines do not contain any of the cells or tissues characteristic of leaves, and conversely, leaves lack all features characteristic of spines*”. He also notes that location is the evidence of the ontogenetic origin of spines as leaves: “*The leaf-nature of spines is certainly understandable from the point of view of location: spine primordia look just like leaf primordia and are produced at a location where we would expect leaf primordia—at the base of the axillary bud’s shoot apical meristem*”.

Based on the location of the microleaf-like structures and the developing spines we observed, we propose that an alternative explanation worth exploring is that these spines may also represent modified shoots. However, our intent is not to challenge existing hypotheses, but to emphasize that *Carnegiea gigantea* seedlings present a valuable system for studying aerial organ primordia. The use of seedlings opens opportunities for molecular techniques, such as in situ hybridization, which can track the expression of regulatory genes in developing organs. By applying modern methods to study aerial organs in the SAM of saguaro seedlings, we can revisit and further investigate with a fresh view the long-standing question of the identity of cactus spines and leaves, contributing to the ongoing exploration of cactus modifications.

## Figures and Tables

**Figure 1 plants-13-03406-f001:**
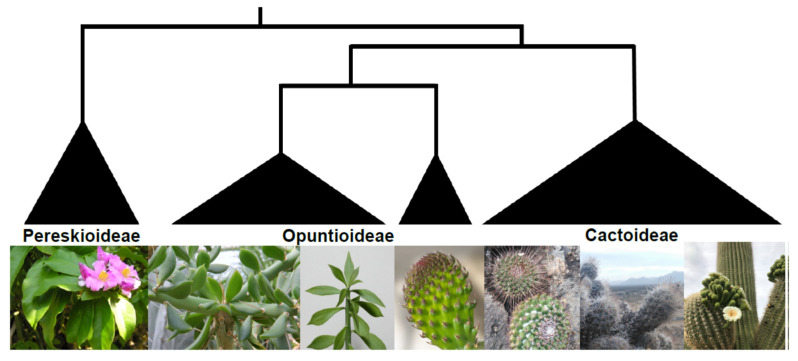
Reduction of leaves through the evolution of major Cactaceae lineages. Early diverging members in the Pereskioideae tribe have broad laminated photosynthetically functional leaves. Opuntioideae tribe members can have photosynthetic succulent leaves to ephemeral foliar buds only present in younger cladodes (e.g., *Opuntia*). In highly succulent members of the Cactoideae tribe, the leaves are reduced to microscopic ‘foliar’ buds in the SAM. Photos from left to right: *Pereskia grandifolia* (D. Boilley), *Quiabentia verticillata* (P.A. Mansfield), *Pereskiopsis* sp. (C.T. Johanson), *Opuntia* sp. (ZooFari), *Mammillaria* sp. (T. Hernández), and *Carnegiea gigantean* (Jrmichae). All pictures except *Mammillaria* sp. were distributed under a CC-BY 2.0 license at Wikimedia Commons.

**Figure 2 plants-13-03406-f002:**
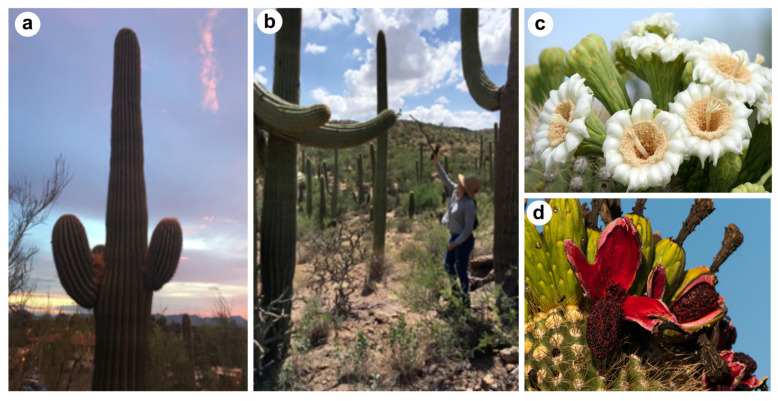
Adult *Carnegiea gigantea* (saguaro) plants and flowers. (**a**,**b**): adult plants of saguaro cactus growing in Tucson, AZ, USA; (**c**) saguaro flower in summer; (**d**) ripe saguaro fruit. Photo credits: (**a**,**b**): Miravel-Gabriel, C and Hernández-Hernández, T; (**c**) L. Hammar and (**d**) R. Cameron. (**c**,**d**) distributed under CC BY-NC-SA 2.0 and CC BY-NC-ND 2.0 licenses respectively, at flickr.com.

**Figure 3 plants-13-03406-f003:**
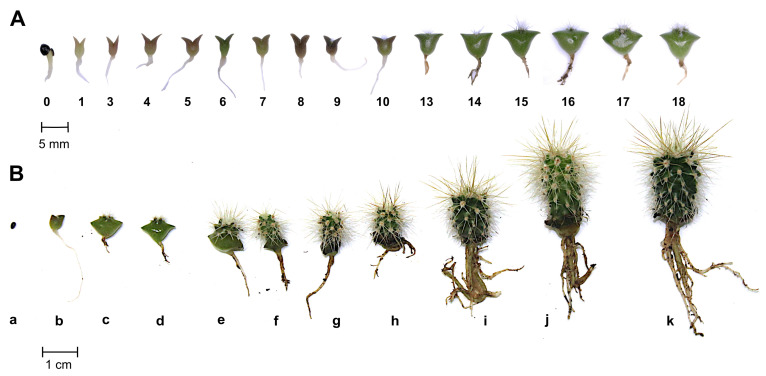
Saguaro seeds germination and seedling growth. In (**A**) days post germination (DPG) over an 18-day period and (**B**) over an 18-month period: (a) seed, (b) 6 DPG, (c) 13 DPG, (d) 16 DPG, (e) 2 months, (f) 3 months, (g) 4 months, (h) 5 months, (i) 8 months, (j) 1 year, and (k) 1.5 years after germination.

**Figure 4 plants-13-03406-f004:**
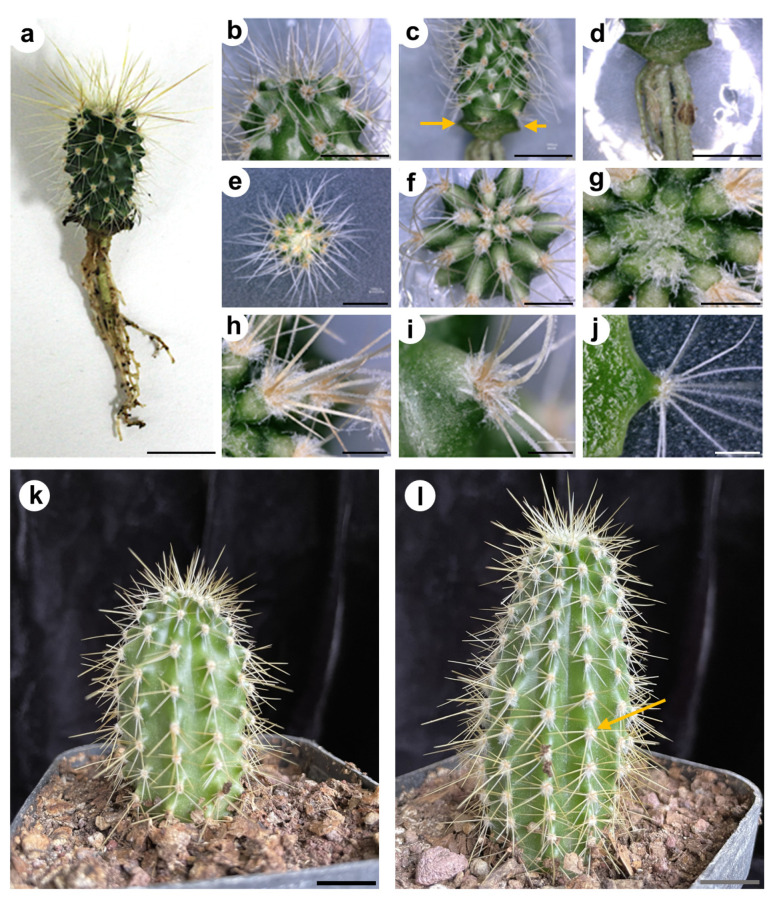
Anatomy and morphology of 1- to 5-year-old saguaro plants. (**a**–**j**) one year old saguaro, (**a**–**d**) side view, (**e**–**g**) top view, (**h**–**j**) tubercle with spines on top. In “a” note the helicoidal arrangement of the first formed tubercles, that will fuse as the young plant keeps developing, to form the longitudinal ribs that characterize a mature saguaro. (**k**,**l**) 4- and 5-year-old saguaros. Yellow arrows in (**c**) indicate cotyledons, yellow arrow in (**l**) indicate the formation of ribs from fused tubercles. Scales bars for (**a**–**d**,**k**,**l**) = 1 cm, (**e**–**g**) = 5 mm, (**h**–**j**) = 1 mm.

**Figure 5 plants-13-03406-f005:**
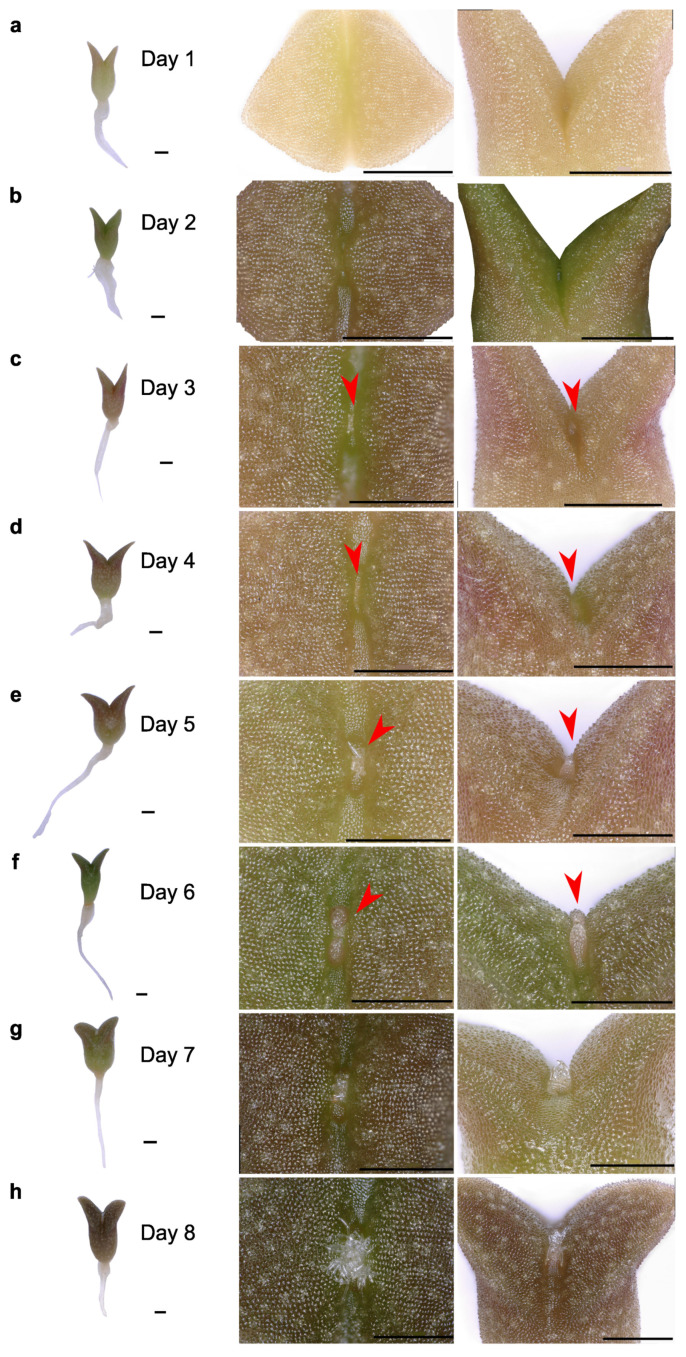
(**a**–**h**) Ontogenetic characterization of saguaro seedling aerial primordia development in the first eight days after germination. Red arrows indicate the first emerged primordia. Scale shown with red bars = 1 mm. (**i**–**p**) Ontogenetic characterization of saguaro seedling aerial primordia development 9 to 17 days after germination. Notice the young tubercles surrounding the center of the SAM. They present spines on their top. As they develop, these tubercles fuse to form ribs. Scale bars = 1 mm.

**Figure 6 plants-13-03406-f006:**
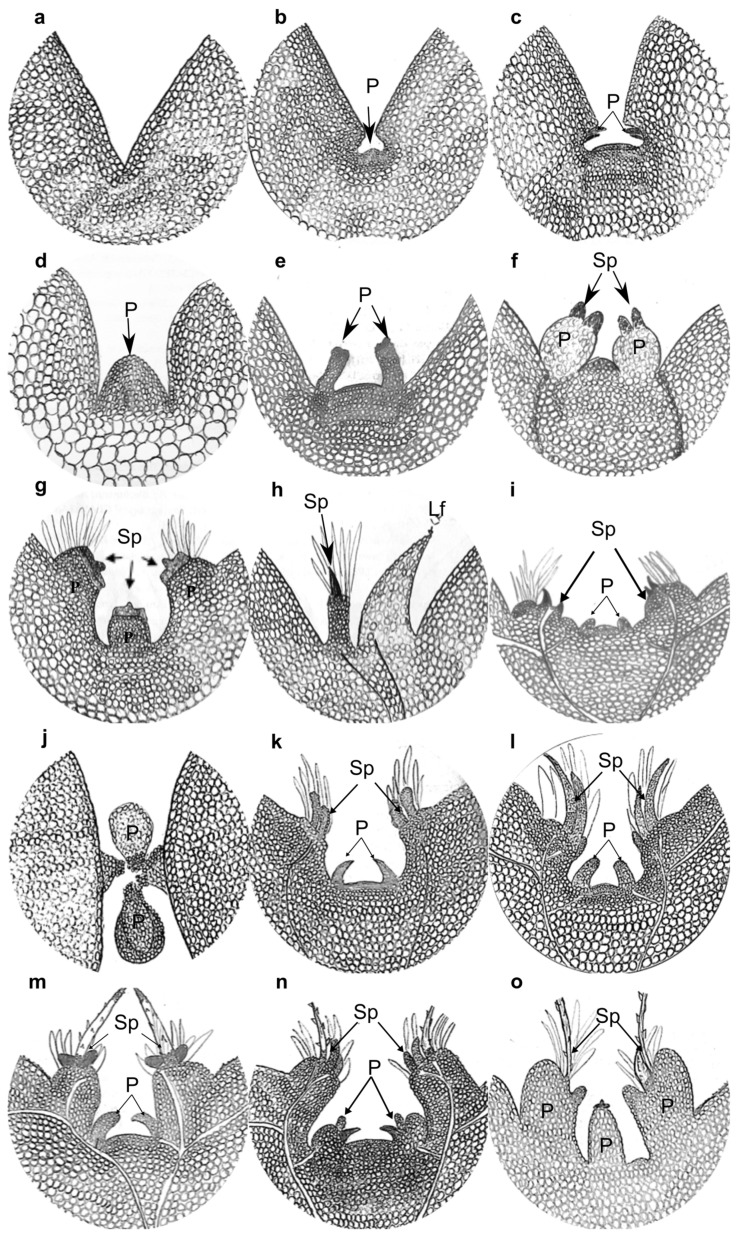
Hand drawings showing a schematic representation of major events during SAM development in saguaro seedlings. (**a**) 1 DPG, no visible SAM; (**b**) 2 DPG, SAM starts to grow and can be seen between the cotyledons, (**c**) 3 DPG; the first two primary primordia (P) are formed parallel to the cotyledon growth axis, (**d**) SAM enlarges, (**e**) 4 DPG, primary primordia enlarge and show abundant meristematic cells at the top, (**f**) 5 DPG; primary primordia increase in size and vascular tissue starts to form underneath them, spine primordia form and enlarge on top of them, (**g**) 7 DPG, three primordia are fully formed already, and trichomes appear at the top of the oldest, (**h**) 8 DPG, in few seedlings we could observe a non-spine, non-trichome primordia structure we refer here as leaf primordia (Lf), (**i**) 9 DPG the first spine primordia structures are seen on top of primary primordia or tubercle primordia, (**j**) 10 DPG, primordia seen from an upper view show an oval-shape, (**k**) spine primordia are clear and start differentiating, (**l**) 12 DPG, vascular tissue is visible enervating the SAM, primordia and cotyledons, (**m**) 13 DPG, clear spines start elongating, and small scales are seen along it, (**n**) 15 DPG, the second set of primary primordia grow and increase in size, (**o**) 17 DPG, the first formed spines are fully formed. P: tubercle or primary primordia; Lf: putative leaf primordia; Sp: Spine primordia.

**Figure 7 plants-13-03406-f007:**
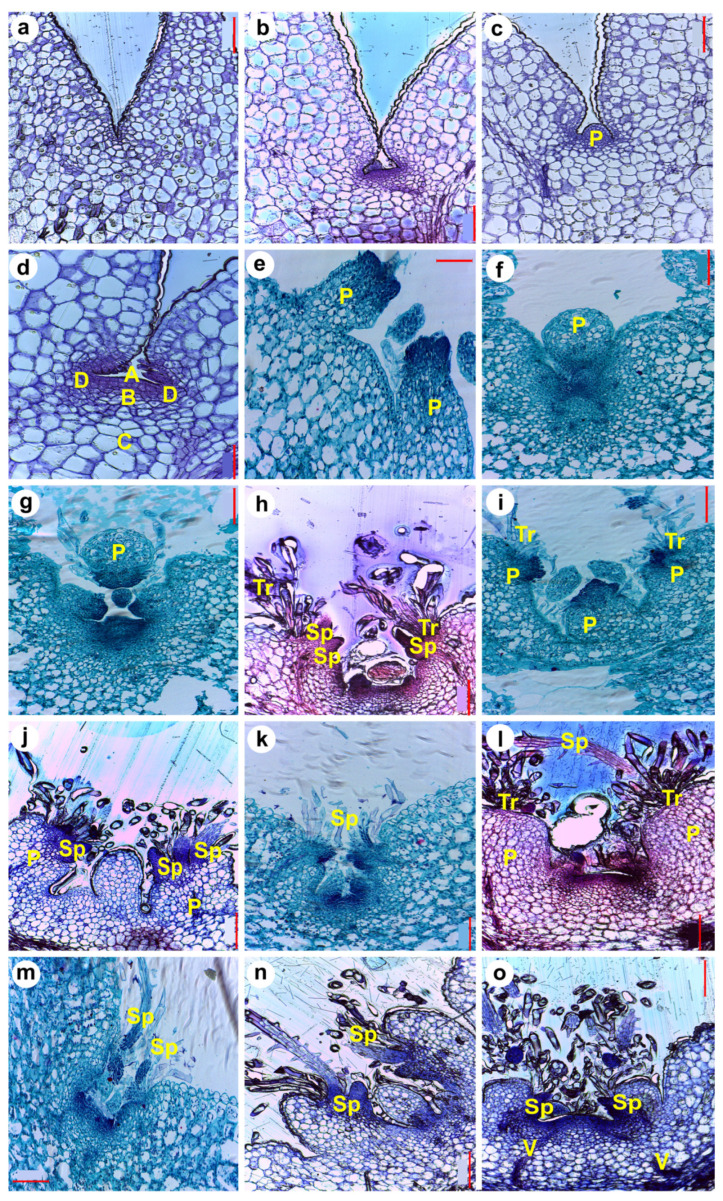
Histological sections showing the development of SAM structures during germination and growth of saguaro seedlings. (**a**) 1 DPG, (**b**) 2 DPG, (**c**) 3 DPG, (**d**,**e**) 4 DPG, (**f**) 5 DPG, (**g**) 6 DPG, (**h**,**i**) 6 DPG, (**j**) 7 DPG, (**k**) 8 DPG, (**l**,**m**) 10 DPG, (**n**) 12 DPG, (**o**) 14 DPG. P = indeterminate primordia, Tr = trichome primordia, Sp = spine primordia, V = vascular tissue, A = tunica, B = central mother cells, C = pith-rib meristem, D = peripheral zone. Scale bars = 0.2 mm, except in (**d**) and (**e**) = 0.1 mm.

**Figure 8 plants-13-03406-f008:**
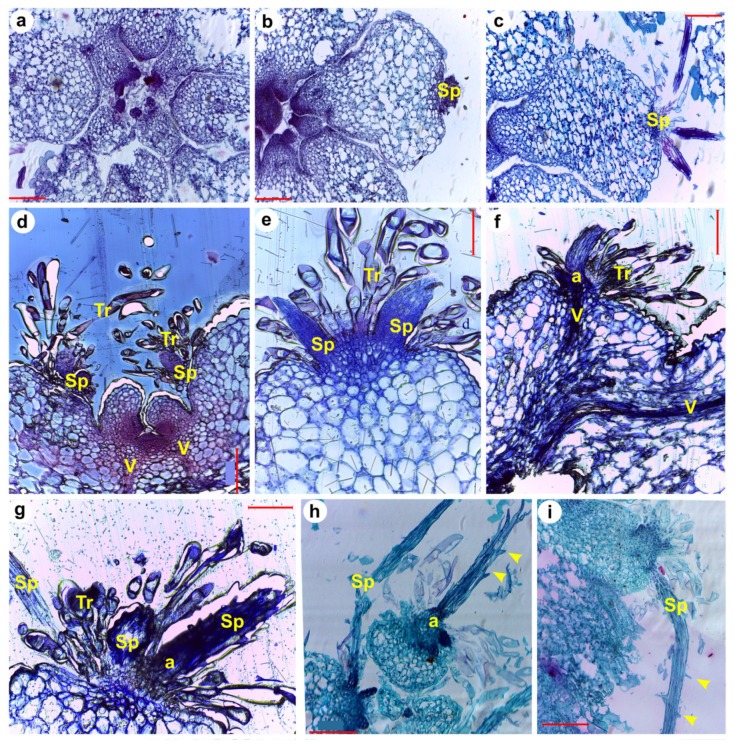
Histological sections showing spine development in saguaro seedlings. (**a**,**b**) 7 DPG showing an early spine primordia development, (**c**) 8 DPG, (**d**) 10 DPG showing vasculature towards the youngest primordia, (**e**) 7 DPG showing trichomes and two spine primordia, (**f**,**g**) 13 DPG, strong staining at the base of the spine suggests high meristematic activity, (**h**,**i**) 10 DPG, with spines that are well developed and start lignifying. Sp = spine primordia, Tr = trichome primordia, V = vascular tissue, a = base of spines, yellow arrowheads indicate trichome-like structures emerging from spines. Scales in red bars = 0.2 mm, except in (**e**) = 0.1 mm.

**Figure 9 plants-13-03406-f009:**
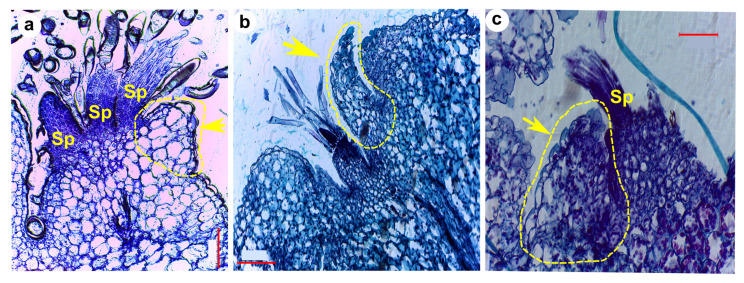
Putative leaf primordia or “microleaf-like” structures observed in the shoot apical meristem (SAM) of saguaro seedlings. (**a**) At 7 DPG, and (**b**,**c**) at 8 DPG. The microleaf-like structures, outlined with yellow dotted lines, exhibit a distinct arrangement and morphology compared to spine primordia (Sp). Arrows indicate regions that may correspond to the abaxial boundary of these structures. While vascular tissue and stomata were not observed, further confirmation through scanning electron microscopy (SEM) is necessary to verify their identity. Scale bars = 0.1 mm.

**Figure 10 plants-13-03406-f010:**
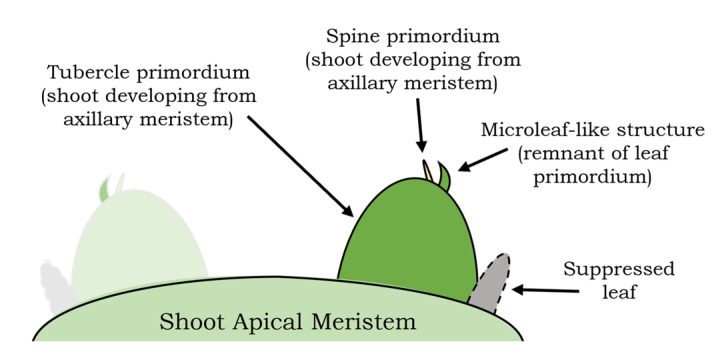
Putative model of the origin of tubercles, spines, and microleaf-like structures in the SAM of the saguaro.

## Data Availability

The original contributions presented in the study are included in the article/Appendix A, further inquiries can be directed to the corresponding authors.

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
