# Peer review of "Are Cactus Spines Modified Leaves? Morphological and Anatomical Characterization of Saguaro Seedlings (Carnegiea gigantea) with Special Focus on Aerial Organ Primordia"

_plants, 2024, doi:10.3390/plants13233406_

Round 1

Reviewer 1 Report

Comments and Suggestions for Authors

The paper aims to introduce seedlings of saruago cactus as a perspective model to study spine development in cacti and their homologies. The paper is nicely illustrated both in the main text and the supplements. I suggest re-organizing the text to get it more concentrated on the suriago seedlings as a model. First of all, I found the title ‘Are cactus spines modified leaves? morphological and anatomical characterization of saguaro (Carnegia gigantea) seedlings with a special focus on aerial organ primordia’ confusing and not reflecting the paper subject. When accepting the review invitation, I expected the paper on spine initiation and their homologies. This is not exactly so. Some other points are listed below.

I believe that in the case of cacti, special characters such as venation or stomata are meaningless in referring spine homologies. The stomata are present both in leaves and stems, and appear quite late in the development. The same is true with vasculature, the pattern of vascular bundle distribution is more important than the presence or absence of the vasculature. The presence of vasculature depends on the organ size, in small and reduced organs it can be absent. For understanding the nature of spines in cacti, the criterion of relative position and the developmental data are more important.

According to the data presented, it is very intriguing that the first two tubercules appear alternating with the cotyledons. If we regard the tubercules as a modified bud, we can expect the tubercle should be initiated in the leaf axil. The first leaves are the cotyledons. But the first pair of tubercles is initiated ‘independently’ of any subtending leaf.

If the tubercle is a secondary order meristem (SAM), then the spines directly ordinated from this meristem should be regarded as modified leaves. If the spines are initiated in the axils of leafy structures, they should be treated as modified shoots. But having in mind the plant construction of two other subfamilies, it is more promising to look for such subtending leaves at the tubercle base (as in Opuntia). Apparently, such leaves are entirely suppressed in Cactiodeae, but they can develop occasionally. The situation is similar to Arabidopsis, where the flower-subtending bracts are absent in the WT. But in some mutants, they are restored.

It would be nice (in the future) to conduct a SEM study in addition to photos under a dissecting microscope and anatomical slides.

Author Response

Dear Reviewer 1,

Thank you for taking the time to provide your thoughtful suggestions and insightful comments. We have carefully followed your recommendations and made the necessary revisions, which we believe have improved the manuscript. We hope that the updated version is now suitable for publication.

Please find our detailed responses below, and we are more than happy to incorporate any additional revisions or clarifications you may suggest.

R1-1. The paper aims to introduce seedlings of saruago cactus as a perspective model to study spine development in cacti and their homologies. The paper is nicely illustrated both in the main text and the supplements. I suggest re-organizing the text to get it more concentrated on the suriago seedlings as a model. First of all, I found the title ‘Are cactus spines modified leaves? morphological and anatomical characterization of saguaro (Carnegia gigantea) seedlings with a special focus on aerial organ primordia’ confusing and not reflecting the paper subject. When accepting the review invitation, I expected the paper on spine initiation and their homologies. This is not exactly so. Some other points are listed below.

Answer.

Thank you for your kind words and thoughtful feedback. We understand your concern regarding the title and appreciate your suggestion. However, after careful consideration, we believe the current title, "Are cactus spines modified leaves? Morphological and anatomical characterization of saguaro (Carnegia gigantea) seedlings with a special focus on aerial organ primordia," accurately reflects the scope and intent of our study.

Our motivation stemmed from an extensive review of the literature regarding the ontogenetic origin of spines in Cactaceae. Although it may challenge an idea that has been accepted for a long time, we believe there is still room for debate as to whether this assumption is fully supported by empirical evidence. As mentioned in our manuscript, there is no unequivocal evidence that the spines in Cactaceae are truly modified leaves (in contrast to alternative explanations such as shoots). From a very early stage in development, spines never exhibit any foliar structure. By establishing saguaroas a model system, we aim to investigate leaf and spine development at genetic and molecular levels, providing additional evidence to clarify the modified ontogenetic pathways in Cactaceae.

Given the importance of your comment, we have enriched and improved the Discussion section, including a speculative idea with a figure about the potential ontogenetic origin of the different aerial organs, such as spines, and now have proposed a putative model. Therefore, if you agree, it may be suitable to retain the original question posed in the title.

In retaining our original title, we seek to reignite this discussion within the context of our detailed study on Carnegia gigantea seedlings and emphasize the relevance of the topic. We believe this perspective is not only pertinent to our research but also significant to the broader field of cactus development. That being said, we are open to modifying the title if both you and the editor feel that it is critical for the paper’s clarity and impact. We are more than willing to make adjustments to ensure the manuscript meets the highest standards.

R1-2. I believe that in the case of cacti, special characters such as venation or stomata are meaningless in referring spine homologies. The stomata are present both in leaves and stems, and appear quite late in the development. The same is true with vasculature, the pattern of vascular bundle distribution is more important than the presence or absence of the vasculature. The presence of vasculature depends on the organ size, in small and reduced organs it can be absent. For understanding the nature of spines in cacti, the criterion of relative position and the developmental data are more important.

Answer. Thank you for your valuable insight. We have incorporated your suggestion into the introduction section on page 4, line 145, where we address the importance of relative position and developmental data in understanding spine homology in cacti.

R1-3. According to the data presented, it is very intriguing that the first two tubercules appear alternating with the cotyledons. If we regard the tubercules as a modified bud, we can expect the tubercle should be initiated in the leaf axil. The first leaves are the cotyledons. But the first pair of tubercles is initiated ‘independently’ of any subtending leaf.

Answer. Thank you for this very interesting observation. Indeed, we were surprised to find this organization of tubercle disposition. At this point we consider it is hard to define whether the tubercle is a modified bud. We even started speculating whether the tubercle could be a modified leaf, which fuses with previously developed ones to form a single column along the stem, or a modified shoot that develops from the “axillary meristem” of a suppressed leaf. Further research may help to explore different possible origins of the tubercle. We have added text in the discussion to include the ideas you proposed.

R1-4. If the tubercle is a secondary order meristem (SAM), then the spines directly ordinated from this meristem should be regarded as modified leaves. If the spines are initiated in the axils of leafy structures, they should be treated as modified shoots. But having in mind the plant construction of two other subfamilies, it is more promising to look for such subtending leaves at the tubercle base (as in Opuntia). Apparently, such leaves are entirely suppressed in Cactiodeae, but they can develop occasionally. The situation is similar to Arabidopsis, where the flower-subtending bracts are absent in the WT. But in some mutants, they are restored.

Answer. Thanks for this also very interesting observation. Yes, if tubercles were a secondary SAM or axillary meristem, spines could be regarded as the modified leaves originating from it. And, yes, if spines would originate from the axils of “leaf” structures, then they would correspond to modified shoots. In most cases, we did not observe spines originating from the axils of leaf-like or other type of structure, at least not in the histology sections or the digital microscopy observations performed in young seedlings. However, in some cases, microleaf-like structures were observed next to the spines.

And, indeed, in Arabidopsis, the “axillary meristem” becomes the floral meristem from which all the floral organs originate, while the original “leaf”, the bract, become suppressed. It is possible that in Saguaro, the “leaf” is suppressed. We agree and have added these ideas in the discussion. We are thankful for your comments, that have helped to significantly improve this section.

R1-5. It would be nice (in the future) to conduct a SEM study in addition to photos under a dissecting microscope and anatomical slides.

Answer. Thank you for the suggestion. It would be indeed revealing to observe these structures in a SEM.

Reviewer 2 Report

Comments and Suggestions for Authors

Dear Authors,

The reviewed work addresses a very important problem of evolution in the Cactaceae family. It is worth noting that the authors have developed useful protocols and procedures for germinating saguaro seeds in laboratory conditions.

Unfortunately the photographic documentation is of very poor quality. Microscopic preparations indicate that the material for research was poorly fixed, there are numerous artifacts. Therefore I cannot accept this m-s in its current form.

Author Response

R2- The reviewed work addresses a very important problem of evolution in the Cactaceae family. It is worth noting that the authors have developed useful protocols and procedures for germinating saguaro seeds in laboratory conditions.

Unfortunately the photographic documentation is of very poor quality. Microscopic preparations indicate that the material for research was poorly fixed, there are numerous artifacts. Therefore I cannot accept this m-s in its current form.

Dear Reviewer 2,

Thank you for your valuable feedback. We understand your concerns regarding the image quality. The resolution of the figures was reduced during the PDF preparation for review, which may have impacted the clarity of the images. However, the original high-resolution images will be included in the final version.

We would also like to emphasize that working with this species has been particularly challenging due to the unique tissue composition and structure of succulents. Extensive optimization was required for the resin embedding and sectioning processes, including prolonged tissue fixation to achieve the best possible conditions. This is quite different from working with model organisms like Arabidopsis, which are more straightforward to handle in histological studies.

Despite these challenges, we consider the histological preparations we are presenting to be of good to excellent quality, and we are confident that the final version of the images will accurately reflect this. We appreciate your understanding and hope the high-resolution images in the final version will meet your expectations.